# Characteristics and outcomes of a cohort of COVID-19 patients in the Province of Reggio Emilia, Italy

**Paolo Giorgi Rossi[1], Massimiliano Marino[2], Debora Formisano[2], Francesco Venturelli [1,3]\*, Massimo Vicentini[1], Roberto Grilli[2], the Reggio Emilia COVID-19 Working Group[¶]**

**1** Epidemiology Unit, Azienda Unità Sanitaria Locale – IRCCS di Reggio Emilia, Reggio Emilia, Italy, **2** Department of Clinical Governance, Azienda Unità Sanitaria Locale – IRCCS di Reggio Emilia, Reggio Emilia, Italy, **3** Clinical and Experimental Medicine PhD Program, University of Modena and Reggio Emilia, Modena, Italy

¶ Membership of the Reggio Emilia COVID-19 Working Group is provided in the Acknowledgments.
* francesco.venturelli@ausl.re.it

**Data Availability Statement:** According to Italian law, anonymized data can only be made publicly available if there is potential for the re-identification of individuals (https://www.garanteprivacy.it).

## Abstract

This is a population-based prospective cohort study on archive data describing the age- and sex-specific prevalence of COVID-19 and its prognostic factors. All 2653 symptomatic patients tested positive for SARS-CoV-2 from February 27 to April 2, 2020 in the Reggio Emilia province, Italy, were included. COVID-19 cumulative incidence, hospitalization and death rates, and adjusted hazard ratios (HR) with 95% confidence interval (95% CI) were calculated according to sociodemographic and clinical characteristics. Females had higher prevalence of infection than males below age 50 (2.61 vs. 1.84 ‰), but lower in older ages (16.49 vs. 20.86 ‰ over age 80). Case fatality rate reached 20.7% in cases with more than 4 weeks follow up. After adjusting for age and comorbidities, men had a higher risk of hospitalization (HR 1.4 95% CI 1.2 to 1.6) and of death (HR 1.6, 95% CI 1.2 to 2.1). Patients over age 80 compared to age < 50 had HR 7.1 (95% CI 5.4 to 9.3) and HR 27.8 (95% CI 12.5 to 61.7) for hospitalization and death, respectively. Immigrants had a higher risk of hospitalization (HR 1.3, 95% CI 0.99 to 1.81) than Italians and a similar risk of death. Risk of hospitalization and of death were higher in patients with heart failure, arrhythmia, dementia, coronary heart disease, diabetes, and hypertension, while COPD increased the risk of hospitalization (HR 1.9, 95% CI 1.4 to 2.5) but not of death (HR 1.1, 95% CI 0.7 to 1.7). Previous use of ACE inhibitors had no effect on risk of death (HR 0.97, 95% CI 0.69 to 1.34). Identified susceptible populations and fragile patients should be considered when setting priorities in public health planning and clinical decision making.

## Introduction

The novel SARS-CoV-2 (COVID-19) pandemic in early 2020 has been threatening the entire word [1, 2]. The virus has shown a high reproduction number and to spread rapidly [3, 4]. Italy has been one of the first countries facing the epidemic outside of China and surely up to

Thus, the data underlying this study are available on request to researchers who meet the criteria for access to confidential data. In order to obtain data, approval must be obtained from the Area Vasta Emilia Nord (AVEN) Ethics Committee, who would then authorize us to provide aggregated or anonymized data. Data access requests should be addressed to the Ethics Committee at CEReggioemilia@ausl.re.it as well as to the authors at the Epidemiology unit of AUSL - IRCCS of Reggio Emilia at info.epi@ausl.re.it, who are the data guardians.

**Funding:** The study has been conducted using institutional funds of the Azienda USL-IRCCS di Reggio Emilia. This study has been partially funded by the Italian Ministry of Health; grant number COVID-2020-12371808. The funder had no role in the definition of the study design neither in the interpretation and publication of the results.

**Competing interests:** The authors have declared that no competing interests exist.

the end of March 2020, it was the most affected Western country [5, 6]. The spectrum of disease of COVID-19 is wide, ranging from no symptoms at all to severe mixed interstitial-alveolar pneumonia often requiring admission in an intensive care unit and ventilation. Fatality rates are high, ranging from 2% to 12%, depending on the country, on reporting systems and definitions, and on length of follow up since disease onset [7]. Further, hospitalization rates change according to different approaches to care and to varying availability of hospital beds; the latter also depends on the place and the phase of the epidemic [7].

Since we are facing a new disease, very few studies can provide information on the factors explaining the variability observed in the fatality rate and on how to predict whether the disease will be severe or not. Therefore, it is hard to define the prognosis both for individuals and for groups of patients. Age and sex seem to be the only confirmed and well described prognostic factors, with a higher case fatality rate in older subjects and in males [8, 9]. Pre-existing chronic conditions have been generically reported as poor prognosis determinants, but the strength of the association between each specific comorbidity and outcomes has not yet been fully explored [10, 11].

Indeed, gaining a better understanding of the role of the main prognostic factors and quantifying the strength of their association with the rate of occurrence of a critical event is essential to identifying patients at high risk of worsening clinical conditions and to assessing the actual needs of different patient groups.

In this report, based on the cohort of all residents in the province of Reggio Emilia who were SARS-CoV-2-positive at nasal and pharyngeal swab and with symptoms (COVID-19 cases) since the inception of the epidemic, we describe patient characteristics and explore their role as putative prognostic factors in predicting the occurrence of hospital admission or death.

## Material and methods

### Study design

This is a population-based prospective cohort study on archive data.

### Setting

The province of Reggio Emilia, located in Northern Italy, has a population of 532,000. Hospital, outpatient, primary, and preventive care to all the resident population is provided by the Local Health Authority, the local public organizational entity of the National Health Service.

The first case of SARS-CoV-2 disease (COVID-19) in the province was diagnosed on February 27, 2020. Up to April 8, there were 3264 confirmed cases in the province; the epidemic was still spreading, but at a lower rate, and cumulative incidence reached about 6 per 1000.

All schools were closed throughout the province on February 22, and some restrictions were placed on social activities. On March 8, strict control measures limiting people's mobility and a partial lockdown was put in place; on March 11, the lockdown was extended, and only essential work activities were allowed.

### Study population

The cohort of COVID-19 patients includes all symptomatic patients who tested positive with PCR between February 27 and April 2, 2020. During the evolution of the epidemic, criteria for testing changed; at an earlier stage (until March 3), all suspected cases with flu-like symptoms, fever, cough, dyspnea, and those who had had a contact with a case or had been in one of the red zones (where the initial cluster occurred) were tested. In this phase, according to the above-mentioned criteria, asymptomatic close contacts of a positive case were also tested. In

the subsequent phase, all those with symptoms suggestive of COVID-19 were tested, regardless of whether not they had had any contact with a positive case, while asymptomatic contacts were no longer tested at all. Access to testing for symptomatic patients was possible through emergency room presentation or through prescription by the public health services or general practitioner, usually by phone. Swabs were performed either at the individual's home or in dedicated clinics.

Since the criteria for testing asymptomatic contacts changed over time, they are excluded from the present cohort.

## Data sources

In the Province of Reggio Emilia, data on patients found positive to SARS-CoV-2 are registered in a special database with a dedicated software made available for the management of each individual case in order to allow epidemiological interviews, contact tracing and surveillance of symptoms through daily phone calls. This dataset registers the date of symptom onset and, for patients in home quarantine, the evolution of symptoms over time hospitalization and death.

This SARS-CoV-2 database was linked with the routinely available administrative databases of the Local Health Authority, which include data for each resident in the Province, in addition to demographic information, hospital discharge abstract data, coded according to the International Classification of Diseases-9-CM (ICD-9-CM) of diagnosis and procedure, and admission and discharge dates, vital status at discharge, and outpatient pharmacy data at the individual prescription level. Data were anonymized, and record linkage procedures were performed according to the unique identification number which is assigned to each resident.

Analysis of previous hospitalizations (up to preceding 10 years), as registered in the local administrative databases, made it possible to identify each individual patients' comorbidities; data on drugs prescribed were also used to identify patients with diabetes and chronic obstructive pulmonary disease (COPD). (S1 File).

## Outcome measures

The outcomes were hospitalization and death. Time to event variable started from symptom inception. Events occurring until April 3, 2020, were included.

## Putative prognostic factors

We considered the following patient characteristics: age, sex, place of birth (Italy or abroad), time span (in days) from symptom onset to diagnosis/ hospitalization, and comorbidities, whose prognostic role was explored both singly (chronic obstructive pulmonary disease, arrhythmia, diabetes, coronary heart disease, heart failure, vascular diseases, obesity) and by computing the Charlson Comorbidity Index, which provides an overall measure of an individual patient's complexity [12]. In particular, we categorized the index in four classes, ranging from 0 (no presence of relevant comorbidity) to $\geq 3$ (indicating the highest level of complexity, in terms of number and/or severity of comorbidities).

Given the current concerns on their possible impact on the clinical evolution of the disease [11], we also evaluated exposure to ACE inhibitors, a class of drugs targeting molecules involved in COVID-19 infection process, and their possible substitute therapy, AT1-antagonists.

## Statistical analyses

Case cumulative incidence and case fatality rates (CFR) in the source population of residents in the Province were estimated both overall and by sex and by age.

Descriptive analyses of patients included in the cohort and rates of hospitalization and death according to the presence of each putative prognostic factor are reported.

Age- and sex-adjusted hazard ratios (HR) with 95% confidence intervals (95% CI) for each putative prognostic factor were estimated for hospitalization and death through multivariate proportional hazard models on time from symptom onset to event. In particular, a first multivariate model was fitted separately for hospitalization and death, including age, sex, Charlson Index, and place of birth as covariates. Then, in order to estimate the actual association between different types of comorbidities with the events of interest, a second model was used that included with the already mentioned covariates the individual comorbidities instead of the Charlson Index. In all the multivariate models we included time from symptom onset to diagnosis (assumed to be a proxy of severity of the disease, as worse-off patients seek medical assistance quicker) and calendar week of diagnosis, both because a variation in patient characteristics over time was observed (Table 2) and because healthcare services experienced different degrees of difficulty in the clinical and organizational management of patients over the weeks due to the different stages of development of the epidemic.

Lastly, in order to assess the influence of individual comorbidities on the rate of occurrence of the outcomes of interest, multivariate proportional hazard models were used for each comorbidity, which was included as covariate in the model along with age and sex.

Multivariate analyses exclude all the patients for whom relevant information was not available. However, excluded cases always represented less than 25% of the whole cohort.

We do not report formal test of hypothesis and p-values with predefined threshold.

Statistical analysis was performed with Stata 13.0 statistical package.

## Ethics approval

The study was approved by the Area Vasta Emilia Nord Ethic Committee on 07/04/2020 n° 2020/0045199.

## Patient consent

In accordance with the Italian privacy law, no patient or parental consent is required for large retrospective population-based studies approved by the competent Ethics Committee if data are published only in aggregated form.

## Results

During the study period, 4551 symptomatic individuals were tested for SARS-CoV-2 infection. The cohort includes 2653 COVID-19 patients, representing all the resident symptomatic patients found positive at RT-PCR from February 27 to April 2, 2020 (Table 1). Overall positivity to test was 58%; it was lower in younger patients (49% in males and 44% in females) than in older patients (66% in males and 64% in females) (Table 1). Positivity decreased with the progression of the epidemic, from 52.7% (106/201) in the first period to 31.8% (375/1179) in the last (Table 2). The mean age was 63.2 and the median time from symptom onset to diagnosis was 4 days, ranging from 0 to 61 days. Males and females were equally represented in the cohort. Age and sex distribution of cases changed during the epidemic (Table 2).

**Table 1. COVID-19 patients characteristics.** COVID-19 cumulative incidence, hospitalizations, and death rates at the population level in the Province of Reggio Emilia, overall and according to age and sex.

| | Population[1] | | Tested for SARS-CoV-2 | | Subjects in the COVID-19 Cohort | | | | Hospitalized | | | | Deaths | | | |
|---|---|---|---|---|---|---|---|---|---|---|---|---|---|---|---|---|
| | Male | Female | Male | Female | Male | | Female | | Male | | Female | | Male | | Female | |
| | N | N | N | N | N | Risk* 1000 | N | Risk* 1000 | N | Risk* 1000 | N | Risk* 1000 | N | Risk* 1000 | N | Risk* 1000 |
| **Total** | 261563 | 270328 | 2140 | 2411 | 1328 | 5.08 | 1325 | 4.90 | 657 | 2.51 | 418 | 1.55 | 143 | 0.55 | 74 | 0.00 |
| **Age** | | | | | | | | | | | | | | | | |
| **< 51** | 160443 | 153270 | 600 | 909 | 296 | 1.84 | 400 | 2.61 | 61 | 0.38 | 46 | 0.30 | 1 | 0.01 | 1 | 0.01 |
| **51–60** | 38645 | 39442 | 407 | 456 | 268 | 6.93 | 260 | 6.59 | 90 | 2.33 | 38 | 0.96 | 6 | 0.16 | 1 | 0.03 |
| **61–70** | 28561 | 31468 | 365 | 253 | 253 | 8.86 | 160 | 5.08 | 144 | 5.04 | 61 | 1.94 | 18 | 0.63 | 5 | 0.16 |
| **71–80** | 21738 | 25402 | 383 | 257 | 257 | 11.82 | 163 | 6.42 | 181 | 8.33 | 110 | 4.33 | 42 | 1.93 | 12 | 0.47 |
| **≥ 81** | 12176 | 20746 | 385 | 536 | 254 | 20.86 | 342 | 16.49 | 181 | 14.87 | 163 | 7.86 | 76 | 6.24 | 55 | 2.65 |

[1]: Number of residents in the Province as of December 31, 2019.

## Estimates of disease cumulative incidence and hospitalization/death rates in the source population

Age and sex distribution of the COVID-19 cohort are in relation to the whole population of residents in the province in order to draw estimates at the population level of disease prevalence and rates of the events of interest. As shown, females were more represented at younger ages (≤ 50 years) and at very old age (≥ 80 years), where women are also much more represented in the general population, while males were more represented between ages 60 and 79. Age-specific risks of disease were higher in males than in females, except for below age 51.

**Table 2. Case fatality rate.** Case fatality rate (CFR) by sex for calendar period of diagnosis.

| Calendar period of diagnosis | | Tested for SARS-CoV-2 | Subjects in the COVID-19 Cohort * | | Age | Hospitalizations | | Deaths | |
|---|---|---|---|---|---|---|---|---|---|
| | | N | N | % per period | Mean | N | % on exposed | N | % on exposed |
| *from 22/2 to 8/3* | Male | 110 | 64 | 60.4% | 64.05 | 46 | 71.9% | 16 | 25.0% |
| | Female | 91 | 42 | 39.6% | 56.76 | 19 | 45.2% | 6 | 14.3% |
| | Overall | 201 | 106 | | | 65 | 61.3% | 22 | 20.8% |
| *from 9/3 to 15/3* | Male | 363 | 242 | 56.7% | 63.17 | 145 | 59.9% | 40 | 16.5% |
| | Female | 352 | 185 | 43.3% | 57.59 | 79 | 42.7% | 15 | 8.1% |
| | Overall | 715 | 427 | | | 224 | 52.5% | 55 | 12.9% |
| *from 16/3 to 22/3* | Male | 617 | 500 | 55.3% | 62.80 | 259 | 51.8% | 72 | 14.4% |
| | Female | 555 | 404 | 44.7% | 60.72 | 159 | 39.4% | 32 | 7.9% |
| | Overall | 1172 | 904 | | | 418 | 46.2% | 104 | 11.5% |
| *from 23/3 to 29/3* | Male | 565 | 363 | 43.9% | 63.21 | 165 | 45.5% | 14 | 3.9% |
| | Female | 719 | 464 | 56.1% | 62.89 | 129 | 27.8% | 18 | 3.9% |
| | Overall | 1284 | 827 | | | 294 | 35.6% | 32 | 3.9% |
| *from 30/3 to 1/4* | Male | 485 | 150 | 40.0% | 63.21 | 35 | 23.3% | 0 | 0.0% |
| | Female | 694 | 225 | 60.0% | 62.89 | 27 | 12.0% | 1 | 0.4% |
| | Overall | 1179 | 375 | | | 62 | 16.5% | 1 | 0.3% |
| Total | | 4551 | 2639 | | | 1063 | 40.3% | 214 | 8.1% |

*For 14 patients the period of diagnosis could not be assessed with sufficient precision.

Age-specific risks of hospitalization and death were higher in males than in females by a factor of 2 or more.

## Overall case fatality rate and rate of hospital admissions

After a median follow up of 14 days, 1075 (40%) and 217 (8.2%) COVID-19 cases experienced hospitalization or death, respectively. The rates of both these events were higher in males than in females (50% vs 31% for hospital admission, and 11% vs 6% for death). For patients followed up for at least four weeks, hospital admission reached 61.3% and death 20.8% (Table 2).

## COVID-19 patient characteristics and rates of critical events

The prevalence of individual characteristics are outlined in Table 3, along with the crude rate of hospital admissions and death for each patient group. The frequency of both outcome measures was related to sex, age, and overall patient complexity as defined by the Charlson Index. Comorbidities were more common in males (72% Charlson Index = 0) than in females (76% Charlson Index = 0). As for single comorbidities (the most prevalent being hypertension, cancer, and diabetes), all were associated with high (i.e. above 50%) rates of hospitalization and death (except obesity, above 15%).

## HRs for hospitalization and death

Results of the multivariate analysis are reported in Table 4 and confirm the association between sex, age, and Charlson Index with both the outcome measures. Immigration status (as represented by place of birth) was found to be associated with hospitalization, with patients born abroad having a 40% higher risk. Longer time span from symptom onset to diagnosis had a lower risk of hospitalization and death, thus confirming that a shorter length of that interval indicates worse clinical condition. Although not statistically significant, HRs for calendar periods of diagnosis suggest a trend towards better outcomes for patients diagnosed in the second part of the study period (i.e. after the third week) compared to those diagnosed in the early phase of the first three weeks of the epidemic.

## Effect of single comorbidities on the risk of hospitalization and death

As shown in Table 5, COPD, chronic kidney disease, and heart failure had the strongest association with the risk of hospitalization, adjusting for age and sex. As for the use of AT-1 inhibitors and ACE inhibitors, exposure to these drugs appeared to be associated with a modest increase in hospitalization risk which, for ACE inhibitors, was not compatible with a random fluctuation. However, this association disappeared when limiting the analysis to the subgroup of patients with coronary heart disease, hypertension, or heart failure.

   The highest risk of death was seen in patients with cardiovascular comorbidities (heart failure, arrhythmia, coronary heart disease), followed by dementia and diabetes. Use of AT-1 inhibitors or ACE inhibitors was not associated with the risk of death.

## Discussion

### Principal findings

Below age 50, females had a higher risk of COVID-19 than did males, but in all other age groups the risk was higher in males. Hospitalization reached 60% and case fatality rate 20% in

**Table 3. Hospitalization and death rates.** Characteristics of COVID-19 cases, hospitalizations, and deaths for each included putative prognostic factor.

| | Total | | Hospitalized | | Deaths | |
|---|---|---|---|---|---|---|
| | N | % of exposure in the population | N | % (out of those exposed) | N | % (out of those exposed) |
| **Total** | 2653 | | 1075 | 40.5% | 217 | 8.2% |
| **Sex** | | | | | | |
| Male | 1328 | 50.1% | 657 | 49.5% | 143 | 10.8% |
| Female | 1325 | 49.9% | 418 | 31.5% | 74 | 5.6% |
| **Age** | | | | | | |
| < 51 | 696 | 26.2% | 107 | 15.4% | 2 | 0.3% |
| 51–60 | 528 | 19.9% | 128 | 24.2% | 7 | 1.3% |
| 61–70 | 413 | 15.6% | 205 | 49.6% | 23 | 5.6% |
| 71–80 | 420 | 15.8% | 291 | 69.3% | 54 | 12.9% |
| ≥ 81 | 596 | 22.5% | 344 | 57.7% | 131 | 22.0% |
| **Time from symptoms to diagnosis** | | | | | | |
| < 5 days | 1407 | 53.7% | 480 | 34.1% | 134 | 9.5% |
| ≥ 5 days | 1212 | 46.3% | 574 | 47.4% | 75 | 6.2% |
| unknown | 34 | | 21 | | 8 | |
| **Place of birth** | | | | | | |
| Italy | 2259 | 91.8% | 997 | 44.1% | 211 | 9.3% |
| Abroad | 202 | 8.2% | 64 | 31.7% | 6 | 3.0% |
| Unknown | 192 | | 14 | | 0 | |
| **Charlson Comorbidity Index** | | | | | | |
| 0 | 1757 | 73.8% | 623 | 35.5% | 89 | 5.1% |
| 1 | 225 | 9.5% | 120 | 53.3% | 34 | 15.1% |
| 2 | 193 | 8.1% | 109 | 56.5% | 27 | 14.0% |
| ≥ 3 | 206 | 8.7% | 143 | 69.4% | 57 | 27.7% |
| Unknown | 272 | | 80 | | 10 | |
| **Single Comorbidities** | | | | | | |
| COPD | 128 | 5.4% | 91 | 71.1% | 24 | 18.8% |
| Coronary heart disease | 168 | 7.1% | 115 | 68.5% | 41 | 24.4% |
| Dementia | 107 | 4.5% | 50 | 46.7% | 25 | 23.4% |
| Diabetes | 284 | 12.0% | 185 | 65.1% | 51 | 18.0% |
| Chronic kidney disease | 59 | 2.5% | 45 | 76.3% | 15 | 25.4% |
| Cancers | 301 | 12.7% | 167 | 55.5% | 44 | 14.6% |
| Hypertension | 430 | 18.1% | 280 | 65.1% | 87 | 20.2% |
| Obesity | 65 | 2.7% | 34 | 52.3% | 8 | 12.3% |
| Heart failure | 137 | 5.8% | 96 | 70.1% | 43 | 31.4% |
| Arrhythmia | 185 | 7.8% | 123 | 66.5% | 46 | 24.9% |
| Dyslipidemia | 118 | 5.0% | 85 | 72.0% | 26 | 22.0% |
| Vascular disease | 61 | 2.6% | 37 | 60.7% | 10 | 16.4% |
| **Use of drugs in previous year** | | | | | | |
| ACE inhibitors | 450 | 17.0% | 277 | 61.6% | 56 | 12.4% |
| AT1 antagonists | 368 | 13.9% | 224 | 60.9% | 52 | 14.1% |

patients with at least four weeks of follow up. We confirm better prognosis for women, a strong effect of age (stronger in males than in females), and worse prognosis for immigrants and for patients with heart failure, arrhythmia, dementia, coronary heart disease, diabetes or hypertension but not for patients with COPD.

**Table 4. Hospitalization and death risk by sociodemographic characteristics and comorbidities index.** Effect of sex, age, calendar period, time from symptom to diagnosis, place of birth, and comorbidities. Models on hospitalization include 1866 patients and 757 outcomes; models on deaths include 2025 patients and 195 deaths.

| | Hospitalization | | Death | |
|---|---|---|---|---|
| | **HR** | **95% CI** | **HR** | **95% CI** |
| **Sex** | | | | |
| Females | 1 | | 1 | |
| Males | 1.4 | (1.2–1.6) | 1.6 | (1.2–2.1) |
| **Age** | | | | |
| < 51 | 1 | | 1 | |
| 51–60 | 1.3 | (1.0–1.8) | 1.5 | (0.5–4.2) |
| *61–70* | 3.2 | (2.4–4.1) | 3.8 | (1.6–9.4) |
| *71–80* | 5.9 | (4.5–7.6) | 9.1 | (4.0–20.6) |
| ≥ 81 | 7.1 | (5.4–9.3) | 27.8 | (12.5–61.7) |
| **Calendar period** | | | | |
| before 15 March 2020 | 1 | | 1 | |
| from 16 to 22 March 2020 | 0.89 | (0.74–1.01) | 1.3 | (0.9–1.8) |
| from 23 to 29 March 2020 | 0.91 | (0.74–1.13) | 0.5 | (0.3–0.8) |
| **Time from symptoms to diagnosis (days)** | | | | |
| *OR per day* | 0.96 | (0.94–0.97) | 0.87 | (0.84–0.90) |
| **Place of birth** | | | | |
| Italy | 1 | | 1 | |
| Abroad | 1.3 | (0.99–1.81) | 1.03 | (0.42–2.56) |
| **Charlson Comorbidity Index** | | | | |
| 0 | 1 | | 1 | |
| 1 | 1.2 | (0.93–1.5) | 1.6 | (1.0–2.5) |
| 2 | 1.6 | (1.2–2.0) | 2.0 | (1.3–3.1) |
| ≥ 3 | 2.1 | (1.6–2.6) | 2.7 | (1.9–3.9) |

## Strengths and weaknesses of the study

The main limitation of this study is that we do not have any information on treatments administered in hospital or prescribed at home. Further analyses, requiring ad hoc data collection, must be conducted to study how therapies interacted with the natural history of the disease and with prognostic factors.

Another limitation of this study is that it is based only on routinely collected hospitalization data to define comorbidities. This source of information clearly underestimates the prevalence of comorbidities that rarely lead to hospitalization, such as obesity, dyslipidaemia, hypertension, or mild COPD. Collecting a long history of hospitalization (as we did, up to 10 years) and integrating it with the use of drugs specific to some chronic conditions (i.e. diabetes, COPD) has been suggested as an effective measure to reduce misclassification and minimize underestimation of the prevalence. On the other hand, using information registered before the onset of the COVID-19 epidemic is the only way to obtain unbiased information on a population-based cohort including non-hospitalized patients. In fact, the probability of registering comorbidities during anamnesis increases with disease severity; this difference in accuracy of exposure ascertainment introduces a bias toward overestimating the impact of comorbidity on prognosis. This bias may be the cause of the high heterogeneity observed in systematic reviews for comorbidities [13, 14].

**Table 5. Hospitalization and risk of death by comorbidities.** Effect of each comorbidity on hazard of hospitalization and death. All hazard ratios are adjusted for age and sex. Models for hospitalizations include 2143 patients and 782 outcomes; models on deaths include 2362 patients and 201 deaths.

| | | Hospitalization | | Death | |
|---|---|---|---|---|---|
| Comorbidities* | | HR | 95% CI | HR | 95% CI |
| | COPD | 1.9 | (1.4–2.5) | 1.1 | (0.7–1.7) |
| | Coronary heart disease | 1.3 | (1.0–1.7) | 1.7 | (1.2–2.5) |
| | Dementia | 1.2 | (0.9–1.8) | 1.8 | (1.1–2.8) |
| | Diabetes | 1.5 | (1.3–1.9) | 1.6 | (1.1–2.2) |
| | Chronic kidney disease | 1.9 | (1.3–2.9) | 1.5 | (0.9–2.6) |
| | Cancers | 1.4 | (1.1–1.7) | 1.4 | (1.0–2.0) |
| | Hypertension | 1.4 | (1.2–1.6) | 1.6 | (1.2–2.1) |
| | Obesity | 1.4 | (0.9–2.0) | 1.3 | (0.6–2.9) |
| | Heart failure | 1.6 | (1.2–2.1) | 2.3 | (1.6–3.2) |
| | Arrhythmia | 1.5 | (1.2–1.9) | 1.8 | (1.3–2.5) |
| | Dyslipidaemia | 1.3 | (0.99–1.69) | 1.4 | (0.9–2.2) |
| | Vascular disease | 1.2 | (0.8–1.8) | 1.2 | (0.6–2.2) |
| *Use of drugs in previous year*** | | | | | |
| | ACE inhibitors | 1.3 | (1.1–1.5) | 0.97 | (0.69–1.34) |
| | AT1 antagonists | 1.2 | (1.0–1.5) | 1.16 | (0.83–1.64) |
| *Use of drugs in previous year**** | | | | | |
| | ACE inhibitors | 1.12 | (0.82–1.54) | 0.8 | (0.50–1.3) |
| | AT1 antagonists | 1.07 | (0.78–1.49) | 1.1 | (0.7–1.8) |

* adjusted for age and sex.

**adjusted for age and sex and Charlson Comorbidity Index.

***Restricted to subjects with at least one of the following comorbidities: coronary heart disease, hypertension, or heart failure; model on hospitalization includes 425 patients and 246 hospitalizations; model on death includes 528 patients and 106 deaths. Adjusted for age and sex and Charlson Comorbidity Index.

We adopted the case definition used by WHO and the Italian Ministry of Health in which only cases positive to RT-PCR SARS-CoV-2 test are considered COVID-19-confirmed cased. Unfortunately, referral to SARS-CoV-2 testing was not standardized and strongly depended on the availability of human and technical resources to collect swabs and perform tests, but also on the awareness of symptoms and on the accessibility of clinics for testing. During the study period, access to testing for paucisymptomatic patients was strongly limited by the lack of human resources to collect swabs at the patient's home, and COVID-19-dedicated clinics outside of the emergency rooms were set up only in the last week of the study period. The absence of uniform criteria for test referral and the context-dependent availability of testing limit the comparability of results between different studies [15–17]. Including only hospitalized patients does not increase comparability, since the availability of hospital resource also changed during the epidemic and from country to country. Further, it introduces a collider bias, as our results suggest, since some comorbidities influence both the probability of death and of being hospitalized. Thus, restricting the population to only hospitalized people may hide the effect of a such comorbidities [18].

## Comparison with other studies and interpretation

While in this study we focused on the risk of hospitalization and death in a cohort of COVID-19 patients diagnosed during the epidemic in Northern Italy, it also provided us with the opportunity to describe the pattern of distribution of the disease in the whole population. We

observed different age-specific risks for females and males resulting in an overall equal proportion of cases. This observation is consistent with previous studies including all symptomatic cases [8, 19, 20] except for a report on the early phases of the epidemic in Lombardy [6]. Indeed, females had a higher risk among people below age 50, while males had higher risk in older ages. The cause of this difference is unknown, but both biological reasons, including hormonal factors in women in reproductive age, and different access to testing should be investigated. Indeed, we observed a higher probability of being tested below age 50 years in women than in men. Surprisingly, we noted a different sex ratio among cases in different phases of the epidemic, with a higher proportion of males at the beginning yet the opposite in the later period under study. This phenomenon, which is unexpected and difficult to explain, could also justify the difference between our study and the report from Lombardy, which was conducted in a much earlier phase of the epidemic.

Consistently with previous findings [9, 10, 19, 21–24], while the risk of disease is approximately similar, the clinical condition seems to be more severe in males than in females.

We confirm the increased risk with age, which remains extremely high even when adjusting for all others characteristic [9, 10, 19, 21–24]. The effect of age is stronger for hospitalization and particularly for death than it is for infection and for males rather than females (Table 1).

Hospitalization and case fatality rates were extremely high in this population-based cohort, reaching 60% and 20%, respectively, in those patients with at least four weeks of follow up. Even if most studies are reporting a case fatality rate of between 1% and 10% [8, 10, 25], cohort studies with sufficient follow up showed similar results [26, 27]. The high fatality rate in our study and in similar studies assessing the prognosis of cases diagnosed during the peak of the epidemic was also due to the limited access to the SARS-CoV-2 test, resulting in the identification of severe cases only.

A previously never-reported finding is the higher hospitalization rate of foreign-born residents than of Italians. We previously reported a similar prevalence of positivity and similar probability of testing between the two groups [28]. This finding is surprising because immigrants, particularly when their arrival in the host country is relatively recent (as is the case in Italy), are usually healthier than native populations and they usually show lower hospitalization and mortality rates [29, 30]. Nevertheless, we could adjust for comorbidities, thus reducing the possible confounding due to the healthy migrant effect. Given that excess risk is appreciable only for hospitalization and not for mortality, it is possible that this is due to the difficulty in effective home quarantine for these patients. Finally, considering that most of the countries of origin have a high prevalence of tuberculosis and BCG is thus recommended, our data do not support the hypothesis that the previously observed non-specific protective effect of BCG on other viral infections [31] is also protective against SARS-CoV-2 infection.

We also found an interesting trend towards a reduced rate of hospitalization and death over the weeks of the epidemic, taking into account patients' age, sex, comorbidities, and length of follow up. While not explained by differences in patient characteristics, the positive trend observed for the two outcome measures considered could, to some extent at least, represent the effect of health professionals and health services rapidly developing the experience required to better cope with the challenges of the clinical and organizational management of a new disease after the first couple of weeks. Nevertheless, as mentioned above, over the 5 weeks representing the time span of this study we saw an increase in diagnoses among females in the last two weeks of the period under study that was not compatible with a random fluctuation, while in the first three weeks we observed more males. This suggests that some underlying characteristics of the case mix may change during the epidemic as the result of changes in the epidemiology of the disease or of changes in the resources available for testing people with less severe symptoms.

Interestingly, in terms of the comorbidities examined, we found an increased risk of hospitalization for COPD but a very small effect on death. This is not consistent with what was reported in a previous study with small numbers [26].

We confirm an important role of several comorbidities, particularly for heart diseases. In general, comorbidities had a stronger association with mortality than with hospitalization, with the only exception being chronic kidney disease. The strongest effects were for heart failure, arrhythmia, dementia, coronary heart disease, diabetes, and hypertension, all with $\geq 50\%$ excess hazard. These data are consistent with recent systematic reviews on the role of cardiovascular diseases and diabetes [13, 14].

Lastly, we did not find evidence of any effect of the use of AT-1 antagonists and ACE inhibitors on hospitalization and death, a reassuring finding that will hopefully be confirmed by others. While an association emerged between ACE inhibitors and hospitalization, it was likely due to residual confounding as it was not confirmed when the comparison between users vs non-users of this drug was performed only among the subgroup of patients with cardiovascular comorbidity. Surprisingly, we found small or no effect for vascular diseases. This is a quite heterogeneous group of diseases and it is possible that we are missing some important prognostic factor due to this grouping, but numbers did not allow for any further distinction. As assessment of obesity and dyslipidaemia through hospital discharge records is challenging, and in our case resulted in an underestimation of the exposure compared to the known prevalence in the general population, the HR that we obtained should be considered carefully [32, 33].

The mechanisms underlying these associations are mostly unknown. A deeper understanding of the causal chain from infection, disease onset, and immune response to outcomes could lead to an explanation of how these prognostic factors act. Nevertheless, quantifying the strength of association between pre-existing conditions and COVID-19 outcomes is important to understand the disease.

## Supporting information

**S1 File.**
(DOC)

## Acknowledgments

¶ The following are members of the Reggio Emilia COVID-19 Working Group:

Massimo Costantini, Giulio Formoso, Emanuela Bedeschi, Cinzia Perilli, Elisabetta Larosa, Eufemia Bisaccia, Ivano Venturi, Cinzia Campari, Francesco Gioia, Serena Broccoli, Marta Ottone, Pierpaolo Pattacini, Giulia Besutti, Valentina Iotti, Lucia Spaggiari, Pamela Mancuso, Andrea Nitrosi, Marco Foracchia, Rossana Colla, Alessandro Zerbini, Marco Massari, Anna Maria Ferrari, Mirco Pinotti, Nicola Facciolongo, Ivana Lattuada, Laura Trabucco, Stefano De Pietri, Giorgio Francesco Danelli, Laura Albertazzi, Enrica Bellesia, Simone Canovi, Mattia Corradini, Tommaso Fasano, Elena Magnani, Annalisa Pilia, Alessandra Polese, Silvia Storchi Incerti, Piera Zaldini, Efrem Bonelli, Bonanno Orsola, Matteo Revelli, Carlo Salvarani. We would like to thank Jacqueline M. Costa for the English language editing.

## Author Contributions

**Conceptualization:** Paolo Giorgi Rossi, Massimo Vicentini, Roberto Grilli.

**Data curation:** Paolo Giorgi Rossi, Massimiliano Marino, Debora Formisano, Francesco Venturelli, Massimo Vicentini, Roberto Grilli.

**Formal analysis:** Massimiliano Marino, Debora Formisano.

**Methodology:** Paolo Giorgi Rossi, Massimiliano Marino, Roberto Grilli.

**Supervision:** Paolo Giorgi Rossi, Roberto Grilli.

**Validation:** Paolo Giorgi Rossi, Massimiliano Marino, Debora Formisano, Francesco Venturelli, Massimo Vicentini, Roberto Grilli.

**Visualization:** Paolo Giorgi Rossi, Massimiliano Marino, Debora Formisano, Francesco Venturelli, Massimo Vicentini, Roberto Grilli.

**Writing – original draft:** Paolo Giorgi Rossi, Massimiliano Marino, Debora Formisano, Francesco Venturelli, Massimo Vicentini, Roberto Grilli.

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
