## [Decision Letter · Decision Letter 0]

24 Jul 2020

PONE-D-20-13898

Characteristics and outcomes of a cohort of SARS-CoV-2 patients in the Province of Reggio Emilia, Italy

PLOS ONE

Dear Dr. Venturelli

Thank you for submitting your manuscript to PLOS ONE. After careful consideration, we feel that it has merit but does not fully meet PLOS ONE’s publication criteria as it currently stands. Therefore, we invite you to submit a revised version of the manuscript that addresses all the points raised during the review process.

We look forward to receiving your revised manuscript.

Kind regards,

Gianluigi Forloni

Academic Editor

PLOS ONE

Journal Requirements:

Reviewers' comments:

Reviewer's Responses to Questions

**Comments to the Author**

1. Is the manuscript technically sound, and do the data support the conclusions?

Reviewer #1: Yes

Reviewer #2: Yes

2. Has the statistical analysis been performed appropriately and rigorously? 

Reviewer #1: Yes

Reviewer #2: I Don't Know

3. Have the authors made all data underlying the findings in their manuscript fully available?

Reviewer #1: Yes

Reviewer #2: Yes

4. Is the manuscript presented in an intelligible fashion and written in standard English?

Reviewer #1: Yes

Reviewer #2: Yes

5. Review Comments to the Author

Reviewer #1: In this Ms., Paolo Giorgi Rossi et al., reported a cohort of SARS-CoV-2 patients in Reggio Emilia. A total of 2653 symptomatic patients were included. They found that patients with heart failure, arrhythmia, dementia, ischemic heart disease, diabetes, and hypertensions show higher risk of hospitalization and of death. ACE inhibitors has no effect on risk of death. This Ms. was not well written. The novel findings in current study should be highlighted. The comparison with previous studies and corresponding interpretations should be more intensive.

Reviewer #2: The manuscript presents data on COVID 19 from patients in the Reggio Emilia Province. Data from symptomatic SARS COV 2 positive patients are set in relation with the routine database of the local health authority. Compared to other data, e.g. from china, Case fatality reported in this study is comparatively high. The manuscript is straight forward and well written. I therefore have only few comments:

1. A major restriction of the study is, that mainly (this changed during the period) symptomatic patients were tested and only symptomatic Sars Cov 2 patients were included in the study. It is therefore impossible to deduct the “prevalence of infection” (e.g. abstract p.2 l.23, discussion p13 l 216) as only the prevalence of symptomatic infection is assessed. This must be changed throughout the manuscript. As a consequence, the data in table 1 (which is not Sars Cov 2 prevalence but prevalence of symptomatic COVID-19) has to be interpreted with caution. The prevalence is 10x higher in >81yrs males compared to <51ys. Is this because risk of infection is higher in elderly? Or just the rate of symptomatic patients in elderly? Or a higher rate of testing? This issue needs more attention in the manuscript.

2. Linked with 1, it is not clearly stated how “symptomatic” was defined and how the physicians decided who was being tested (as only epidemiological data is used). If the authors could give more information here, this would certainly strengthen the manuscript.

3. Prevalence of comorbidities is deducted from data on previous hospitalizations (routine database). As far as I understand, someone who was never hospitalized during the past 10 years and has e.g. hypertension or COPD managed in outpatient treatment would not be counted as a patient with comorbidity. This weakness is already discussed on p14 l227ff. In my view, at least for SARS positive patients – and for those the increased risk for hospitalization and death connected with pre-conditions was mainly calculated – pre-conditions should have been assessed during the “epidemiological interviews” (p6 l90) or certainly during hospitalization for COVID and should have been included in the “special database” of the cohort. Has this been done? If not it has to be clearly stated.

6. PLOS authors have the option to publish the peer review history of their article (what does this mean?). If published, this will include your full peer review and any attached files.

Reviewer #1: No

Reviewer #2: No

---

## [Author Response · Author response to Decision Letter 0]

10 Aug 2020

RE: Done

RE: The differences in age and sex distribution of cases over the weeks of the epidemic are reported in Table 2. Regarding the proportion of people without hospitalizations reporting comorbidities, the data are reported in the text only and not in a table, so it is not true that “data [were] not shown”. Indeed, there was no reason to say, “data not shown”, sorry. 

Funding: we previously stated that the study did not receive any external funding but in July 2020 we were notified that a proposal for a grant submitted by our group to the Italian Ministry of Health on this topic has been accepted (grant number COVID-2020-12371808). This grant will cover the publication costs and partially the cost sustained for collecting data. We added this in the Funding section. 

5. Review Comments to the Author

Reviewer #1: In this Ms., Paolo Giorgi Rossi et al., reported a cohort of SARS-CoV-2 patients in Reggio Emilia. A total of 2653 symptomatic patients were included. They found that patients with heart failure, arrhythmia, dementia, ischemic heart disease, diabetes, and hypertensions show higher risk of hospitalization and of death. ACE inhibitors has no effect on risk of death. This Ms. was not well written. The novel findings in current study should be highlighted. The comparison with previous studies and corresponding interpretations should be more intensive.

RE: We have revised the Discussion and updated the references. We have focused on methodological issues that could explain the heterogeneity in results found in different studies. 

Reviewer #2: The manuscript presents data on COVID 19 from patients in the Reggio Emilia Province. Data from symptomatic SARS COV 2 positive patients are set in relation with the routine database of the local health authority. Compared to other data, e.g. from china, Case fatality reported in this study is comparatively high. The manuscript is straight forward and well written. I therefore have only few comments:

RE: We thank the reviewer for his encouraging comments. 

1. A major restriction of the study is, that mainly (this changed during the period) symptomatic patients were tested and only symptomatic Sars Cov 2 patients were included in the study. It is therefore impossible to deduct the “prevalence of infection” (e.g. abstract p.2 l.23, discussion p13 l 216) as only the prevalence of symptomatic infection is assessed. This must be changed throughout the manuscript. As a consequence, the data in table 1 (which is not Sars Cov 2 prevalence but prevalence of symptomatic COVID-19) has to be interpreted with caution. The prevalence is 10x higher in >81yrs males compared to <51ys. Is this because risk of infection is higher in elderly? Or just the rate of symptomatic patients in elderly? Or a higher rate of testing? This issue needs more attention in the manuscript.

RE: We agree with the reviewer that this is a limitation of our study; we cannot determine whether differences in disease occurrence were due to differences in the access to testing, to differences in the probability of infection, or to differences in the probability of having symptoms, once infection occurred. We are presenting data on cumulative incidence of COVID-19 diagnosed and confirmed through RT-PCR SARS-CoV-2 test according to the case definition adopted by the Italian Ministry of Health. To be consistent, we have substituted SARS-CoV-2 with COVID-19 in the manuscript where appropriate.

We have added a better description of how the swabs were taken, including a column in Table 1 reporting the number of swabs taken, by sex and age, and in Table 2 by epidemic period. This information helps when interpreting differences by age, showing that older people, at least in this phase of the epidemic, were probably tested when symptoms were more predictive of COVID-19 than were younger patients. This information highlights the population-based nature of the study. Furthermore, we have added a sentence in the limitations section of the discussion.

2. Linked with 1, it is not clearly stated how “symptomatic” was defined and how the physicians decided who was being tested (as only epidemiological data is used). If the authors could give more information here, this would certainly strengthen the manuscript.

RE: We reported in the text the exact list of symptoms adopted by the Italian Ministry of Health to guide field investigations during the epidemic. We have added a paragraph explaining how people had access to the test, which is a more context-specific information. 

3. Prevalence of comorbidities is deducted from data on previous hospitalizations (routine database). As far as I understand, someone who was never hospitalized during the past 10 years and has e.g. hypertension or COPD managed in outpatient treatment would not be counted as a patient with comorbidity. This weakness is already discussed on p14 l227ff. In my view, at least for SARS positive patients – and for those the increased risk for hospitalization and death connected with pre-conditions was mainly calculated – pre-conditions should have been assessed during the “epidemiological interviews” (p6 l90) or certainly during hospitalization for COVID and should have been included in the “special database” of the cohort. Has this been done? If not it has to be clearly stated.

RE: During the epidemiological interviews, collection of comorbidities was not structured; this information may be present in a free text field used to report special situations, but reporting was surely not systematic. Further, while comorbidities are routinely collected during emergency room visits, but without using a structured form, reporting is once again not systematic. Finally, during hospitalization, the anamnesis is more systematically collected, but as the structured medical records are ward-specific, reporting is not uniform anyway. What we noted is that the probability of reporting comorbidity increased with the severity of disease and with the intensity of the care setting. The result is that the measure of comorbidities as a prognostic factor can be done only if we use a source of information that is independent of and possibly acquired before the COVID-19 diagnosis.

---

## [Editor Report · Decision Letter 1]

14 Aug 2020

Characteristics and outcomes of a cohort of COVID-19 patients in the Province of Reggio Emilia, Italy

PONE-D-20-13898R1

Dear Dr. Venturelli,

We’re pleased to inform you that your manuscript has been judged scientifically suitable for publication and will be formally accepted for publication once it meets all outstanding technical requirements.

Kind regards,

Gianluigi Forloni

Academic Editor

PLOS ONE
---

## [Editor Report · Acceptance letter]

18 Aug 2020

PONE-D-20-13898R1 

Characteristics and outcomes of a cohort of COVID-19 patients in the Province of Reggio Emilia, Italy 

Dear Dr. Venturelli:

I'm pleased to inform you that your manuscript has been deemed suitable for publication in PLOS ONE. Congratulations! Your manuscript is now with our production department. 

Kind regards, 

on behalf of

Dr. Gianluigi Forloni 

Academic Editor

PLOS ONE